# Targeting Metabolic Dysfunction for the Treatment of Mood Disorders: Review of the Evidence

**DOI:** 10.3390/life11080819

**Published:** 2021-08-11

**Authors:** Brett D. M. Jones, Salman Farooqui, Stefan Kloiber, Muhammad Omair Husain, Benoit H. Mulsant, Muhammad Ishrat Husain

**Affiliations:** 1Department of Psychiatry, University of Toronto, Toronto, ON M5T 1R8, Canada; Brett.jones@mail.utoronto.ca (B.D.M.J.); stefan.kloiber@camh.ca (S.K.); Omair.Husain@camh.ca (M.O.H.); Benoit.Mulsant@utoronto.ca (B.H.M.); 2Centre for Addiction and Mental Health, Toronto, ON M6J 1H4, Canada; 3Department of Biology, University of Ottawa, Ottawa, ON K1N 6N5, Canada; sifarooqui01@gmail.com

**Keywords:** major depressive disorder, bipolar disorder, metabolic function

## Abstract

Major depressive disorder (MDD) and bipolar disorder (BD) are often chronic with many patients not responding to available treatments. As these mood disorders are frequently associated with metabolic dysfunction, there has been increased interest in novel treatments that would target metabolic pathways. The objectives of this scoping review were to synthesize evidence on the impact on mood symptoms of lipid lowering agents and anti-diabetics drugs, while also reviewing current knowledge on the association between mood disorders and dyslipidemia or hyperglycemia. We propose that metabolic dysfunction is prevalent in both MDD and BD and it may contribute to the development of these disorders through a variety of pathophysiological processes including inflammation, brain structural changes, hormonal alterations, neurotransmitter disruptions, alteration on brain cholesterol, central insulin resistance, and changes in gut microbiota. Current evidence is conflicting on the use of statins, polyunsaturated fatty acids, thiazolidinediones, glucagon-like peptide agonists, metformin, or insulin for the treatment of MDD and BD. Given the paucity of high-quality randomized controlled trials, additional studies are needed before any of these medications can be repurposed in routine clinical practice. Future trials need to enrich patient recruitment, include evaluations of mechanism of action, and explore differential effects on specific symptom domains such as anhedonia, suicidality, and cognition.

## 1. Introduction

Major depressive disorder (MDD) and bipolar disorder (BD) are a leading cause of disability worldwide and result in substantial impairment in functioning for impacted individuals [1]. Current pharmacotherapies for mood disorders include antidepressants, mood-stabilizers, and antipsychotics. Although a beneficial option for many patients, a substantial portion of patients do not respond or reach remission [2,3]. Since many pharmacological approaches for mood disorders target similar biological mechanisms, such as monoamine neurotransmission, it is important to develop novel and repurposed pharmacotherapies that target differential pathological targets that may be implicated in mood disorders.

Given the high prevalence of treatment resistance in mood disorders, investigating the repurposing of medications targeting novel biological targets is necessary. The potential for the use of repurposed medication has been shown in previous work that has attempted to target putative pathological processes associated with mood disorders, such as an activated immune system [4]. Similarly, there has been increasing interest in exploring altered metabolic function in mood disorders as a potential treatment target. In recent years, several studies have evaluated metabolic modulating agents, such as lipid lowering agents and anti-diabetic drugs, for the treatment of mood disorders. The objective of this review is to summarize evidence for associations between mood disorders and metabolic dysfunction (i.e., dyslipidemia and hyperglycemia.) and to provide a scoping review on evidence from clinical trials of lipid lowering agents and anti-diabetics drugs for the treatment of mood disorders.

## 2. Materials and Methods

Relevant literature related to the topic of the current review was obtained by searching Medline using PubMed for reviews and primary studies of the associations between metabolic function and mood disorders (MDD or BD) using the following search terms (depress* or bipolar) and (metabolic function or metabolic dysfunction). Reference lists of related reviews were searched for additional studies. To identify clinical trials of metabolic modulating agents as monotherapy or augmentation treatments in MDD or BD, the following search terms were used: (antidiabet* OR lipid OR hyperglycemic OR statin OR insulin OR thiazolidinedione* OR glucagon-like peptide OR GLP OR metformin OR Poly Unsaturated Fatty Acid OR PUFA OR eicosapentaenoic acid EPA OR DHA OR Docosahexaenoic acid or omega-3 OR omega-6) and (depress* or bipolar) and (trial OR RCT)).

Studies selected were randomised controlled trials (RCTs), cluster RCTs, or cross-over trials.

When a significant number of RCTs were available for one specific class of metabolic-targeting medication, we reviewed recent meta-analyses from reputable peer-reviewed journals that followed standard (e.g., PRISMA) reporting guidelines. When no RCT was available, we reviewed open-label trials or cohort studies. Participants had to meet criteria for MDD or BD according to international classification of disease (ICD) ICD 10, diagnostic statistical manual of mental disorders (DSM)-IV or DSM-5 [5,6]. We only reviewed English language studies.

## 3. Results

### 3.1. Evidence for Associations between Metabolic Dysfunction and Mood Disorders

Individuals with MDD have a 4-fold higher rate of early death than healthy controls, in part due to a higher prevalence of metabolic disturbances such as dyslipidemia and hyperglycemia [7,8]. The association between metabolic disturbances and mood disorders appeared to be bidirectional [9]. In longitudinal studies, several metabolic dysregulations have been associated with sustained depressive states over time [10,11,12].One-year prior to diagnosis, patients with BD present with higher rates of hypertension, cardiovascular disease, gastrointestinal disease, respiratory diseases, metabolic disease, and musculoskeletal disease than the general population [13]. The following section will review specific abnormalities in lipid and glucose regulation.

#### 3.1.1. Lipid Regulation

In a large cohort study (N = 766,427), when compared to controls, patients with MDD had both a higher prevalence of hyperlipidemia (14.4% vs. 7.9%, OR: 1.67; 95%CI: 1.53–1.82) and a higher incidence (3.6% vs. 2.6%, RR: 1.35; 95%CI: 1.24–1.47) [14]. In the same cohort, the prevalence of hyperlipidemia for BD was 13.5% (OR: 1.75, 95%CI: 1.52–2.02) and the incidence was 4.37% (RR: 1.66, 95%CI: 1.47–1.87) [15]. In a separate retrospective cohort study (N = 26,852), patients with newly diagnosed hyperlipidemia had a higher risk for MDD compared to healthy controls (HR: 1.64, 95%CI, 1.55–1.74) [16]. In a large cohort of men (N = 29,133) followed for 5–8 years, baseline low cholesterol was associated with a high relative risk of admission to hospital for MDD or death by suicide [17]. Lower levels of cholesterol have also frequently been associated with suicidality [18]. Alexithymia, which is an inability to recognize emotions frequently seen in MDD, has also been associated with lower high-density lipoprotein (HDL) and suicidality [19]. In a study in older adults, baseline low HDL was associated with a higher risk of MDD within 5 years [20]. In 1040 women, the OR of prevalence of MDD or high level of depressive symptoms was 1.5 for those with low HDL; in 752 men, those with low low-density lipoprotein (LDL) had a 2.0 OR of prevalence and subsequent depression [21]. In a population study, compared to patients with schizophrenia, those with BD had an increased OR of hyperlipidemia 1 year prior to receiving their respective psychiatric diagnosis (OR: 1.22 95%CI: 1.09–1.36) [13].

Alterations in polyunsaturated fatty acids (PUFA)- e.g., omega-3 PUFA, omega-6 PUFA, docosahexaenoic acid (DHA), and eicosapentaenoic acid (EPA)- have also been reported in patients with MDD. The rates of MDD are lower in countries with a high fish consumption and those with lower PUFA consumption have higher rates of depressive symptoms [22,23]. Lower fish intake has been associated with higher rates of suicidality and higher prevalence of BD 1 or 2 [24,25]. Patients with MDD have lower omega-3 PUFA levels than healthy controls [26]. Furthermore, patients with MDD have altered ratio of omega-PUFA with higher amounts of omega-6 vs omega-3 PUFA [27]. In a meta-analysis of patients with BD 1 (n = 118) vs healthy controls (n = 147), patients with BD had significant lower levels of DHA and EPA [28]. There may also be a link between low EPA + DHA erythrocyte levels and risk of developing BD depression or mania in persons at high-risk of BD [29]. Taken together, meta-analyses have found, with high heterogeneity, that MDD is associated with increased total cholesterol, triglycerides, low LDL, low HDL, and low omega-3 PUFA [30,31,32,33].

#### 3.1.2. Glucose Regulation

Clinically significant depressive symptoms are prevalent in up to 30% of patients with diabetes mellitus (DM) [34]. The prevalence of MDD in diabetic patients is twice that of non-diabetic patients (OR: 2.0, 95%CI: 1.8–2.2) and can be as high as 10% [35]. In a large meta-analysis, patients with MDD (N = 154,366) had a higher risk of type 2 DM than healthy persons (N = 2,098,063; RR: 1.49; 95%CI: 1.29–1.72) [36]. In a separate longitudinal study, depressive symptoms at baseline were associated with increased incident type 2 DM over a 3-year follow-up period [9]. In a recent study, hyperglycemia but not insulin resistance, irrespective of diagnosis of DM, was associated with an increased risk of developing depressive symptoms over 4 years [37]. In a cohort of 123,232 patients with DM and 1,933,218 controls, women with DM had 2.55 times higher risk of being diagnosed with MDD than women without DM (95%CI: 2.48–2.62) while the increased risk in men was not as marked (OR: 1.85, 95%CI: 1.80–1.91).

As with MDD, a strong association has been found between BD and DM. DM rates have been reported to be up three times as high in patients with BD as in healthy controls [38]. In a Taiwanese population study, prevalence of DM in patients with BD was higher than in the general population (10.77% vs. 5.57%, OR: 2.01; 99% CI: 1.64–2.48) [39]. Similarly, in a Swedish population study BD was more common in patients with DM (OR: 1.71, 95%CI: 1.54–1.91) than in non-diabetic persons [40]. Another study reported than in the 1 year prior to diagnosis of BD, the OR of presence of DM compared to the general population was 1.27 (95%CI: 1.12–1.45) and the 3-year pre diagnosis OR was 1.53 (95%CI: 1.31–1.79) [13].

### 3.2. Evidence on the Efficacy of Metabolic Modulating Agents for the Treatment of Mood Disorders

Based on our search, we screened 3923 articles and identified 1 open-label study, 19 RCTs, and 4 meta-analyses for this review (Table 1 and Table 2). Included medications were statins, PUFA, thiazolidinediones (TZD), glucagon-like peptide 1 (GLP-1) agonists, metformin (MET), and insulin.

#### 3.2.1. Lipid Lowering Agents for Mood Disorders

##### Statins for the Treatment of MDD

Statins are 3-hydroxy-3-methylglutaryl-coenzyme A reductase inhibitors that are primarily used to treat dyslipidemia [41]. The first RCT of statins in mood disorders investigated the efficacy of lovastatin as an augmenting agent of fluoxetine (up to 40 mg/day) [42]. Participants (N = 68) had MDD and were randomized to either fluoxetine + lovastatin (30 mg/day) or fluoxetine + placebo for 6 weeks. Participants receiving lovastatin had a significantly larger reduction in depressive symptoms (mean (SD): −12.8 (6.3) vs. −8.2 (4.0), t = 3.4, df = 60, *p* < 0.001). A second RCT in patients with severe MDD (N = 60) compared the efficacy of citalopram (40 mg/day) + atorvastatin (20 mg/day) vs citalopram (40 mg/day) + placebo for 12 weeks [43]. The atorvastatin group had a significantly larger reduction in depressive symptoms than in the placebo group (F (3, 174) =8.93, *p* < 0.001). A third study in MDD (N = 48) investigated the efficacy of fluoxetine (40 mg/day) + simvastatin (20 mg/day) vs. fluoxetine (40 mg/day) + placebo for 6 weeks. Again, there was a significantly larger reduction in depressive symptoms in the simvastatin group than in the placebo group, (F (1.88, 78.94) = 3.78, *p* = 0.02) [44].

Two RCTs have investigated the efficacy of statins in patients with MDD with co-morbid medical conditions [45,46]. The first was an RCT in patients with MDD (N = 58) who had a coronary-artery bypass graft (CABG) within the past 6 months [45]. Patients were randomized to either simvastatin (20 mg) or atorvastatin (20 mg) monotherapy with no other antidepressant treatment. Both treatment groups had a significant reduction in depressive symptoms; however, there was a larger and faster improvement on depressive symptoms with simvastatin group than with atorvastatin (F (1.62, 71.06) = 3.41, *p* = 0.048). A second study was a retrospective secondary analysis of an RCT of escitalopram (N = 149) vs placebo (N = 151) for the treatment of MDD in patients who had an acute coronary syndrome 2–14 weeks prior [46]. Participants were prescribed statins as medically indicated by a cardiologist (N = 226). In a logistic regression predicting response rates, escitalopram use and not statins use significantly predicted rates of response (defined as 50% reduction in Hamilton depression rating scale (HDRS)) at 24 weeks. Conversely, statin use but not escitalopram uses significantly predicted response rates at 1-year follow-up. Some patients with MDD who did not participate in the escitalopram randomization received statin therapy (N = 97); depression response rates were significantly higher in those who took lipophilic statins (N = 52; 44.7%) than in those who took hydrophilic statins (N = 45; 17.1%) [46]. More recently, a study of youth and young adults with moderate to severe depression investigated the efficacy of rosuvastatin (N = 48), aspirin (N = 40), or placebo (N = 42) in addition to standard antidepressant treatment for patients with moderate to severe MDD over 12 weeks [47]. There was a significant reduction in depressive symptoms in all three groups and no significant differences in the primary or secondary outcomes between groups. In a recent meta-analysis that pooled the results from the aforementioned studies, statins were more efficacious than placebo in addition to antidepressants to reduce depressive symptoms at 8 weeks (N = 255, SMD = −0.48, 95%CI: −0.74 to −0.22) and 12 weeks (N = 134, SMD = −0.47, 95%CI: −0.89 to −0.05); an exploratory network meta-analysis suggested that simvastatin, which is more lipophilic, may be more efficacious than the rosuvastatin, with is less lipophilic [48].

##### Statins for the Treatment of BD

The data from an RCT of the efficacy of atorvastatin (N = 27) vs. placebo (N = 33) in treating lithium-induced diabetes insipidus in BD and MDD were used to explore the effect of atorvastatin on cognition and mood [49]. There were no differences between the two groups on global cognition, cognitive impairment, executive function, depression relapse, or mania relapse [49]. In a small RCT (N = 54), patients with BD in a manic phased received lovastatin in a addition to lithium compared to placebo and lithium [50]. Overall there was no significant difference in reduction in manic symptoms compared to placebo. In a subsequent small RCT (N = 27), patients with BD in a manic phase were randomized to addition of either placebo or lovastatin to standard of care [51]; there were no differences in mood symptoms between the two groups.

##### PUFAs for the Treatment of MDD

Converging evidence suggests that PUFA are anti-inflammatory, hypotriglyceridemic, can modulate lipid metabolism, regulate adipokines, promote adipogenesis, and alter epigenetic mechanisms [52]. In a recent meta-analysis of 26 trials (N = 2160) in MDD or depressive symptoms found that, PUFAs were more effective than placebo in reducing depressive symptoms (SMD= −0.28; *p* = 0.004) [53]. Pure EPA and a mixture of EPA and DHA (with >60% EPA) demonstrated clinical benefits compared to placebo (SMD = −0.50, *p* = 0.003, and SMD = −1.03, *p* = 0.03, respectively), while pure DHA did not exhibit benefits. In another meta-analysis s of RCTs including only participants with MDD and no medical comorbidity (N = 910), PUFAs were superior to placebo in reducing depressive symptoms (SMD: 1.24; 95%CI: 0.060–2.414) [54]. In a network meta-analysis of the same RCTs, high-dose omega-3 PUFA was more effective than low dose omega-3 PUFA [54].

##### PUFA for the Treatment of BD

RCTs have had mixed results for treating either manic or depressive phases of BD. In a meta-analysis of 5 RCTs (N = 291) for BD depression, omega-3 PUFA yielded a moderate effect size (0.34) [55]. However, PUFA were not better than placebo for the treatment of mania [55]. A recent study examined whether PUFA supplementation would prevent mood episodes in patients with BD over 52 weeks [56]. This small study (N = 80) found little benefit for prophylactic supplementation other than a small reduction in hypomanic symptoms in the omega-3 PUFA group. One small RCT (N = 31) explored the effects of omega-3 PUFA on cognition in BD [57], showing no improvement in cognition compared to placebo.

#### 3.2.2. Antidiabetic Agents for the Treatment of Mood Disorders

##### Thiazolidinediones (TZD) for MDD

TZD function as agonists to the proliferator activated nuclear receptor (PPAR)-gamma, which primarily functions to enhance insulin sensitization and enhance glucose metabolism [58]. PPAR-gamma agonists had previously been assessed in the treatment of other neurological disorders, such as Alzheimer’s disease or multiple sclerosis [59,60]. Animal models of MDD and 2 open-label studies in adults with MDD or BD suggested potential benefit of these agents in the treatment of MDD [61,62,63,64,65,66]. A first RCT was completed in 40 patients with MDD and no history of DM or metabolic syndrome [67]. Participants received citalopram (30 mg/day) + pioglitazone (15 mg/day) or citalopram (30 mg/day) + placebo for 6 weeks. The pioglitazone group had significantly larger reduction in depressive symptoms than the placebo group (F (1, 38) =9.483, *p* = 0.004). A subsequent RCT compared the efficacy of pioglitazone and metformin, and assessed whether their antidepressant effects were due to insulin sensitization [68]. Patients had MDD and polycystic ovarian syndrome (PCOS) (N = 50) and were assigned to pioglitazone (15 mg/day) vs metformin (750 mg twice/day). Overall, pioglitazone was superior to metformin in reducing depressive symptoms (38.3% vs. 8.3% reduction respectively; F (1, 37) = 75.513, *p* < 0.001). At the end of the trial, there were no significant differences in biochemical profile or insulin resistance between the 2 groups, and change in insulin resistance was not associated with change in depressive symptoms [68]. A subsequent RCT in patients with MDD who did not achieve remission with standard antidepressants, randomized participants (N = 37) to either pioglitazone (15 mg/day) or placebo [69] and found no difference in reduction of depressive symptoms in the two groups [69]. However, in a post-hoc analysis splitting the participants based on insulin sensitivity, the insulin resistant pioglitazone group had a significantly larger reduction in depressive symptoms than the insulin resistant placebo group. Further, improvement in oral glucose tolerance test (OGTT) and fasting plasma glucose (FPG) was associated with a favourable reduction in depressive symptoms in the insulin resistant group only [69].

##### Thiazolidinediones (TZDs) for BD

In an open-label study in depressed patients with BD (N = 34) receiving pioglitazone (15–30 mg/day), higher baseline IL-6 was associated with a favourable antidepressant response and improvement in insulin resistance [70]. A subsequent 6-week RCT (N = 44) explored whether pioglitazone (30 mg/day) vs placebo would be an effective augmentation agent to lithium in patients with BD I depression. These patients did not have DM or evidence of metabolic dysfunction. The pioglitazone group had a significant larger reduction in depressive symptoms than the placebo group at week 2, 4 and 6 [71]. Another RCT in patients with BD depression (N = 37) evaluated pioglitazone (15–45 mg/day) vs placebo in addition to standard care^72^. It found no significant benefit of pioglitazone augmentation, with a near significant improvement favouring placebo [72]. Of note, participants recruited to this RCT were not enriched for evidence of metabolic dysfunction.

##### Glucagon-like Peptide 1 (GLP-1) Agonists (Liraglutide) for MDD and BD

GLP-1 is a hormone synthesized in the gastro-intestinal tract (GI) and central nervous system (CNS). Liraglutide, a GLP-1 agonist is primarily used in the treatment of type 2 DM through enhanced glucose utilization via increased insulin secretion and suppression of glucagon [73,74,75]. A small open-label study has investigated whether liraglutide, improves cognition in euthymic patients with MDD (N = 13) or BD (N = 6) [76]. Participants presented with executive dysfunction at baseline and received 1.8 mg/day of liraglutide in addition to standard pharmacotherapy for their mood disorder. A significant pre-post improvement in trail-making test-B (TMTB) scores was observed and it was larger in those with higher baseline BMI and insulin resistance.

##### Metformin for MDD

Metformin (MET), primarily used for treatment of type 2 DM, minimizes hepatic glucose output and improves insulin-mediated uptake of glucose [77]. In observational studies, patients with MDD and DM receiving an antidepressant and MET have improvement in both conditions [78]. In an RCT, 58 patients with MDD and type 2 DM were treated with either MET vs placebo for 24 weeks [79]. MET significantly enhanced cognitive functions across multiple domains (i.e., verbal memory, visual memory, attention, and delayed memory). Further, MET significantly decreased depressive symptoms compared to placebo. MET also decreased HBA1c, with a positive correlation between reduction in HbA1c and depressive symptoms in the MET (r = 0.618, n= 22, *p* < 0.01). A recent RCT explored whether MET would be effective as an adjunctive agent in patients with MDD without medical comorbidity. Patients with MDD (N = 80) were randomized to receive fluoxetine (20 mg/day) + MET (1000 mg/day) or fluoxetine (20 mg/day) + placebo [80]. The MET group had a significantly larger reduction in depression severity than the placebo group after 12 weeks of treatment (SMD: −3.454; 95%CI: −4.145–−2.76). The MET group also had a significantly larger reduction in TNF-alpha, IL-1B, IL-6, IGF-1, MDA, CRP, and significantly increased BDNF and serotonin than the placebo group.

##### Insulin for MDD

Insulin is an endogenous hormone secreted by pancreatic-B cells and is used in replacement therapy for both type 1 and 2 DM. In healthy adults, intranasal insulin has been associated with enhanced mood and well-being [81]. In a cross-over RCT, patients with MDD (N = 35) were randomized to either intranasal insulin (40 IU QID) or placebo [82]. Overall, there was no significant difference between groups in changes in cognitive function or in secondary measures such as quality of life.

##### Insulin for BD

In an RCT, adults with euthymic BD (N = 62) were randomized to intranasal insulin (40 IU QID) vs. placebo [83]. Treatment with insulin led to a significant improvement in executive function as measured by the trail making test-B.

## 4. Discussion

This review highlights converging evidence that mood disorders are associated with metabolic dysfunction of lipid, fatty acid, and glucose regulation. This is supported by animal models, large population-based cross-sectional studies, cohort studies, and observational studies. Small studies provide some evidence for the effectiveness of metabolic modulating agents for the treatment of MDD and BD. The best evidence exists for the use of statins and PUFAs in patients with MDD. Some evidence also supports the use of PUFAs in patients with BD depression, but there is only very limited evidence for the use of any metabolic modulating agent for manic phases. Overall, this field of studies is in its early stages and large RCTs are required to replicate and extend these findings before treatment recommendations can be made.

While it is clear that MDD and BD are associated with metabolic dysfunction, and treatments that target these pathways have shown promise in treating depressive symptoms, it is less clear how these disturbances contribute to the pathophysiology of mood symptoms. Several pathophysiological links and shared mechanisms between metabolic dysfunction and mood disorders have been proposed. Some of the proposed pathophysiological links through which metabolic dysfunction may contribute specifically to the development of mood disorders include and potential therapeutic benefit include; Brain Atrophy: Repeated periods of hypoglycemia/hyperglycemia, lacunar infarcts, and generalized microvascular dysfunction have been shown to impact brain function in regions that are associated with mood and cognition such as the pre-frontal cortex (PFC) and hippocampus (HC) [84]. Brain cholesterol metabolism: Brain cholesterol is responsible for regulation of several processes including membrane bound proteins, ion channels, synaptic transmission, synapse formation, dendrite formation, and axonal formation; disruptions of which have been associated with mood disorders [85]. Monoamine neurotransmission: Deficits in fatty acids have been associated with altered DA and serotonin neurotransmission in the frontal cortex, suggesting impacts on cognitive function and emotion evaluation and regulation [26,86,87,88,89,90]. Inflammation: Cholesterol, LDL, very-LDL (VLDL), and chylomicrons circulating in vessel walls can become oxidized, triggering an immunological cascade, reactive oxygen species (ROS), and immune cells such as macrophages, natural killers cells, mast cells, or dendritic cells [91]. Many of these proinflammatory molecules cross the blood brain barrier (BBB) leading to microglial activation and downstream processes that can lead to psychiatric symptoms and potentially mood disorders [4]. Medications that target metabolic variables, such as Omega-3 PUFA and statins, have also been shown to be anti-inflammatory [92]. Insulin Resistance: Insulin receptors are expressed in brain regions that are associated with mood disorders; the nucleus accumbens (NAc), the ventral tegmental area (VTA), the amygdala, and the raphe nuclei of which disrupted signaling may be contributing to depressive symptomatology [93]. Gut Microbiome: There is a rapidly increasing body of research that has shown that regulation of the gut microbiome has impact on psychiatric disorders [94]. The microbiome-brain axis has also been implicated in brain development, function, and metabolism [95]. Metabolic dysfunction such as obesity, glycemic control, and insulin resistance have been shown to be associated with a shift of microbiota [96].

A major limitation to most treatment studies thus far has been the lack of pre-treatment enrichment of samples. Some of the reviewed RCTs have attempted to stratify participants for evidence of metabolic dysfunction, for example patients with MDD after a CABG (who may benefit from lipid lowering agents) or patients with co-morbid DM/insulin resistance (who may benefit from anti-diabetic agents). Following this approach, a proof-of-concept study recruited participants with MDD and baseline inflammation and found that those with higher inflammation responded best to treatment with PUFAs [97]. If replicated, this finding could suggest that the mechanism of action of PUFAs may in part be anti-inflammatory and patients with baseline inflammation may be more likely to benefit from them. Similarly, many of the anti-diabetic agents reviewed were studied in patients with various baseline levels of insulin resistance or hyperglycemia, and changes in depressive symptoms was associated with changes in insulin resistance [69], glucose tolerance tests [69], inflammation [70], or HBA1C [79]. These findings, support the hypothesis that these agents may be most efficacious in patients with higher degree of metabolic dysfunction. Future studies should consider similar stratification approaches to improve our understanding of mechanism of action of these agents and achieve a higher likelihood of detecting significant treatment effect.

As discussed above, overwhelming evidence support a bi-directional association between metabolic dysfunction and mood disorders. However, a biomarker specific for metabolic dysfunction in individuals with, or at risk for, associated with mood disorders remains elusive. Numerous attempts over the past decades have failed to find a singular “biomarker” of pathology in mood disorders. A composite measure based on multiple biomarkers, similar to the validated measures that predict the risk of stroke, myocardial infarction, or death in patients with vascular risk factors, may be able to distinguish individuals with metabolic dysfunction at risk for mood disorders, or individuals with mood disorders at risk for metabolic disorders. A recent study using machine learning sought to determine the metabolic profile of MDD (N = 5283) compared to controls (N = 10,145) based on a panel of lipids, fatty acids, low molecular weight metabolites, and lipid/fatty acid ratios [98]. A profile of ten metabolomic molecules were associated with increased odds of MDD. In a separate analysis in a subset of these patients, patients who were currently depressed had differential metabolic profile compared to healthy controls [99]. These findings support the notion that in a subset of patients there is an association with lipid/metabolomic regulation and prevalence and incidence of MDD that may be best identified through an index combining multiple measures or a profile based on multiple markers. Future research should explore whether stratification of patients based upon such an index or profile would enhance treatment response to metabolic modulating agents and whether changes in metabolic parameters are associated with response.

Limitations: This scoping review may not have identified all the extant literature on the effect of pharmacotherapies that target metabolic dysfunction to ameliorate mood and related symptoms in patients with MDD and BD.

## 5. Conclusions

In summary, there is evidence of altered metabolic function either through lipid metabolism or hyperglycemia/insulin resistance in a subset of patients with MDD and BD. These metabolic disturbances may have both direct and indirect effects on the pathophysiology of MDD and BD. While there is promising evidence for the use of some metabolic agents for the treatment MDD and BD, larger and better designed studies are required before these agents can be repurposed and recommended for the treatment for mood disorders. These future studies also need to explore potential mechanisms of action and use innovative clinical trial design to address the heterogeneous clinical phenotypes of mood disorders.

## Figures and Tables

**Table 1 life-11-00819-t001:** Clinical trials of metabolic modulating agents in Major Depressive Disorder (MDD).

Reference	Participants (N, % Female)	Age (SD) *	Medication	Comparator	Diagnosis	Enrichment	Outcome Measure and Duration	Result
**Statin**								
Ghanizadeh et al. 2013	N = 68, 63.2%	32.5 (10.2) and 31.7 (9.3)	fluoxetine (40 mg) + lovastatin (30 mg)	Fluoxetine (40 mg) + placebo	DSM-IV MDD	none	HDRS, 6 week RCT	Reduction of 12.8 (6.3) vs. 8.2 (4.0) *p* < 0.001 favouring lovastatin
Haghighi et al. 2014	N = 60, 56.6%	33.1 (8.9) and 31.4 (7.8)	citalopram (40 mg) + atorvastatin (20 mg)	citalopram (40 mg) + placebo	DSM-5 MDD	none	HDRS, 12 week RCT	week 12 HDRS atorvastatin: 19.63 (3.16) placebo: 22.03 (3.58)
Gougol et al. 2015	N= 48, 60.4%	36.4 (8.1) and 34.2 (10.8)	fluoxetine (40 mg) + simvastatin (20 mg)	fluoxetine (40 mg/day) + placebo	DSM-IV MDD	none	HDRS, 6 week RCT	group × time interaction favouring simvastatin (F (1.88, 78.94) = 3.78, *p* = 0.02) (Cohen’s d at week 6: 0.61)
Abbasi et al. 2015	N = 46, 32.6%	56.9 (6.9) and 57.7 (7.3)	simvastatin (20 mg/day)	atorvastatin (20 mg/day)	DSM-IV MDD	CABG in the last 6 months	HDRS, 6 week RCT	significant effect for time × treatment interaction favouring simvastatin s (F (1.62, 71.06) = 3.41, *p* = 0.048)
Kim et al. 2015	N = 300, 39.6%	60.3 (10.8), 58.9 (12.9), 60.4 (10.8), 59.3 (10.0)	escitalopram + statins	Escitalopram only, statin only, placebo only	DSM-IV MDD	acute coronary syndrome (ACS)	HDRS, 24-week RCT with 1-year follow-up	statin use associated with higher 1 year response rate 2.23 (1.11–4.51) *p* = 0.025
Berk et al. 2020	N = 130, 60%,	20.2 (2.6)	Rosuvastatin (10 mg)	Aspirin (100 mg) or placebo	DSM-IV MDD	None	MADRS, 12-week RCT	no difference between rosuvastatin and placebo (−4.2, 95%CI (−9.1, 0.6), *p* = 0.089)
Giorgi et al. 2021			Meta-analysis	statins compared to placebo in 5 RCT at 8 weeks (N = 255, SMD = −0.48, 95%CI = −0.74 to −0.22) and 12 weeks (N = 134, SMD = −0.47, 95%CI = −0.89 to −0.05)				
		**Poly Unsaturated Fatty Acid (PUFA)**						
Liao et al. 2019			Meta-analysis	Beneficial effect of omega-3 polyunsaturated fatty acids on depression symptoms in 26 RCTs (SMD = −0.28, *p* = 0.004)				
Luo et al. 2020			Meta-analysis	n-3 PUFAs were superior to placebo in 10 RCTs (SMD: 1.243 ± 0.596; 95%CI: 0.060~2.414) both the high (SMD: 0.908 ± 0.331; 95%CI: 0.262, 1.581) and the low-dose (SMD: 0.601 ± 0.286; 95%CI: 0.034, 1.18) n-3 PUFAs were superior to placebo, and the efficacy of high-dose n-3 PUFAs is superior to that of low-dose				
		**Thiazolidinediones (TZD)**						
Sepanjnia et al. 2012	N = 40, 72.5%	31.4 (5.4) and 32.7 (5.4)	citalopram (30 mg) + pioglitazone (15 mg BID)	citalopram (30 mg) + placebo	DSM-IV MDD	None	HDRS, 6-week RCT	mean difference favouring pioglitazone (−3.4 (−5.6 to −1.2)).
Kashani et al. 2013	N = 50, 100%,	21.2 (3.3) and 20.3 (4.6)	pioglitazone (15 mg BID)	metformin (750 mg BID)	DSM-IV MDD	PCOS	HDRS, 6-week RCT	percentage reduction favouring pioglitazone [38.3% versus 8.3% reduction from baseline scores, F (1, 37) = 73.513, *p* < 0.001].
Lin et al. 2015	N = 37, 78.4%	49.4 (15.1) and 43.3 (11.8)	pioglitazone (30 mg) +TAU	placebo + treatment as usual	DSM-IV MDD	None	HDRS, 12-week RCT	no significant difference in mean decline of HDRS-21 scores between treatment groups (t [29] = −1.22, *p* = 0.23).
		**Glucagon-Like Peptide 1 (GLP-1) Agonists**						
Mansur et al. 2017	N = 19, 57.9%	38.2	liraglutide (1.8 mg) + TAU		DSM 5 MDD or BD	measurable impairment in executive function	TMTB, 4-week open-label	significant improvement in TMTB standard score with those with higher baseline BMI and IR having the best improvement (Cohen’s d = 0.64, *p* = 0.009).
		**Metformin (MET)**						
Guo et al. 2014	N= 58, 37.9%,	54.7 (7.3) and 53.3 (7.3)	Metformin (1–2 g per day)	placebo	DSM-IV MDD	Diabetes Mellitus type 2	wechsler memory scale–revised and MADRS, 24-week RCT	significant improvement in verbal memory index, visual memory index, general memory index, and delayed memory index compared to placebo. metformin significantly decreased MADRS (F 1112 = 26.43; *p* < 0.001 Metformin significantly decreased MADRS compared to placebo (F = 26.43; *p* < 0.001) MET metformin significantly decreased MADRS (F 1112 = 26.43; *p* < 0.00
Abdallah et al. 2020	N = 80, 47.5%	34.1 (8.4) and 35.1 (8.0)	metformin (1 g) + fluoxetine (20 mg)	placebo + fluoxetine 20 mg	DSM-IV MDD	none	HDRS, 12-week RCT	12 week results favouring metformin LSMD −3.454, *p* = 0.000
		**Insulin**						
Cha et al. 2017	N = 35, 62.9%	47.1 (9.9)	intranasal insulin (40 International Units (IU) QID)	placebo	DSM-IV MDD	none	positive and negative affect schedule (PANAS) and global index of neurocognition, 12-week cross over RCT	no between group differences observed

HDRS: hamilton depression rating scale; RCT: Randomized Controlled Trial; SMD: Standard Mean Difference; TMTB: trail-making test- B; CABG: coronary artery bypass graft surgery; BMI: body mass index; IR: insulin resistance; MADRS: Montgomery–Åsberg Depression Rating Scale; TAU: treatment as usual; RCT: Randomized Controlled Trial. * age is reported for active medication group and then comparator group.

**Table 2 life-11-00819-t002:** Clinical trials of metabolic modulating agents in Bipolar Disorder (BD).

Reference	Participants (N, % Female, Age)	Age (SD) *	Medication	Comparator	Diagnosis	Enrichment	Outcome Measure and Trial Duration	Result
**Statin**								
Fotso Soh et al. 2020 (secondary analysis)	N = 60, 58.3%	47.8 (13.8) and 53.1 (11.8)	atorvastatin (20 mg) + lithium	placebo + lithium	DSM-IV BD or MDD	lithium induced diabetes insipidus	global cognition Z-score, 12-week RCT	No between group differences
Lotfi et al. 2017	N = 27	Not reported	Lovastatin +TAU	Placebo + TAU	DSM-IV BD (mania)	none	YMRS, 4-week RCT	No significant difference found between groups
Ghanizadeh et al. 2013	N = 54, 44.4% female	30.5 (8.1) and 29.5 (10.8)	Lovastatin + lithium	Placebo + lithium	DSM-IV BD (mania)	none	YMRS, 4 weeks	No significant difference found between groups
	**Poly Unsaturated Fatty Acid (PUFA)**							
Sarris et al. 2012			5 RCT (N = 291) for bipolar depression revealed a significant effect size (0.34) in favor of omega-3 (*p* = 0.029) 5 RCT (N = 291) for bipolar mania revealed a non-significant effect					
McPhilemy et al. 2020	N = 80, 51.25%	45 (13) and 48 (12)	EPA (1 g) + TAU	Placebo + TAU	DSM-IV BD (any mood state)	3 mood relapses in last 5 years and 2 in the last 3 years	mood state relapse (mania or depression), 52-week RCT	no significant differences in the number of mood episode relapses (U = 490.00, *p* = 0.14) and no significant difference for time to relapse between groups (Log Rank χ^2^ = 0.41, *p* = 0.52).
Ciappolino et al. 2020	N = 31, 71%,	36 (12) and 50.4 (11.3)	DHA supplementation (1250 mg) + TAU	Placebo + TAU	DSM-IV BD (euthymia)	None	BAC-A, 12-week RCT	no significant effects on cognition between groups.
	**Thiazolidinediones (TZD)**							
Zeinoddini et al. 2015	N = 44, 34.1%	33.6 (3.5) and 31.2 (5.6)	pioglitazone (30 mg) + lithium	Placebo + lithium	DSM-IV BD (depressed)	None	HDRS, YMRS, 6-week RCT	significant effect for time × treatment interaction on the HDRS scores [F (2.78, 116.65) = 4.77, *p* = 0.005]
Aftab et al. 2019	N = 38, 64.9%,	46.7 (12.1) and 43.7 (13)	pioglitazone (15–45 mg) + TAU	Placebo + TAU	DSM-IV BD (depressed)	None	IDS-C30, 8-week RCT	mean reduction from baseline to week 8 in IDS-C30 score was−6.59 for pioglitazone and −11.63 for placebo
	**Insulin**							
McIntyre et al. 2012	N = 62, 46.8%,	40.8 (10.2) and 39.3 (10.4)	intranasal insulin (40 IU q.i.d). + TAU	Placebo + TAU	DSM-IV BD (euthymia)	None	CVLT-II and PDT, 8-week RCT	significant improvement versus placebo with intranasal insulin therapy on secondary measure of executive function (i.e., TMTB). No group differences on primary measure

CVLT-II: California Verbal Learning Test, second edition; PDT: Process Dissociation Task; TMTB: Trail-Making test-B; BAC-A: Brief Assessment of Cognition in Affective Disorder; TAU: Treatment as usual; YMRS: Young Mania Rating Scale; HDRS: Hamilton Depression Rating Scale; IDS-C30: Inventory of depressive symptomatology; DHA: docosahexaenoic acid; EPA; eicosapentaenoic acid. * age is reported for active medication group and then comparator group.

## Data Availability

Not applicable.

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
