# Peer review of "Targeting Metabolic Dysfunction for the Treatment of Mood Disorders: Review of the Evidence"

_life, 2021, doi:10.3390/life11080819_

Round 1
Reviewer 1 Report
In this scoping review Jones and colleagues presented an overview of the literature exploring the correlation between metabolic disorders and mood disorders, with a specific focus on the clinical trials where medications commonly used to treat metabolic disorders were used in the psychiatric setting.
The article is well presented and provides a useful reference for the readers.
I have some suggestions and comments, which are reported below:
- the abstract begins with addressing the issue of treatment resistance in mood disorders. This may mislead the readers since treatment resistance is not addressed in this review
- same applies to the beginning of the introduction section. I suggest removing reference to treatment resistance, unless the authors systematically discuss the potential connection between metabolic disorders and resistance to treatment with psychotropic medications
- moreover, authors state that the fact that most of the psychotropic medications used to treat mood disorders address the monoaminergic neurotransmission could largely contribute to explain the mechanisms underling treatment resistance. This is one of the many hypothesis, and is only focused on the pharmacodynamics. I suggest rephrasing or extending this section in the manuscript
- in the abstract, line 11, please remove the word "treatment" as this is a repetition
- in line 18, please list also other mood disorders as reported in DSM, or rephrase, since MDD and BD are the two most common but not the only mood disorders
- in line 19, the word "inflicted" does not sound appropriate
- in line 31, authors define treatment resistance, but only for MDD and not for BD. Please see also my previous comments regarding this issue
- in line 47, authors state that relevant literature was obtained: please define if relevant refers to quality or congruency with the topic
- similarly, in line 57, please describe how high quality was defined for meta-analyses
- in line 185, authors report previous findings for TZD and other disorders, but neurological would sound more proper than neuropsychiatric, since findings only relate to Alzheimer and multiple sclerosis
- in line 294, please report PFC and HC in full
Author Response
Comment: In this scoping review Jones and colleagues presented an overview of the literature exploring the correlation between metabolic disorders and mood disorders, with a specific focus on the clinical trials where medications commonly used to treat metabolic disorders were used in the psychiatric setting.
The article is well presented and provides a useful reference for the readers.
I have some suggestions and comments, which are reported below:
Response: We would like to thank the reviewer for taking the time to review our manuscript and providing their helpful comments.
Comment: the abstract begins with addressing the issue of treatment resistance in mood disorders. This may mislead the readers since treatment resistance is not addressed in this review.
Response: Thank you for this comment. We have revised abstract as suggested by the reviewer and removed reference to treatment resistant mood disorders.
Comment: same applies to the beginning of the introduction section. I suggest removing reference to treatment resistance, unless the authors systematically discuss the potential connection between metabolic disorders and resistance to treatment with psychotropic medications
Response: Thank you for this comment. We have removed the reference that defines treatment resistance so not to mislead the readers. We have maintained two sentences related to treatment resistance as it emphasizes the clinical need for developing novel and repurposed treatments.
Comment: moreover, authors state that the fact that most of the psychotropic medications used to treat mood disorders address the monoaminergic neurotransmission could largely contribute to explain the mechanisms underling treatment resistance. This is one of the many hypothesis, and is only focused on the pharmacodynamics. I suggest rephrasing or extending this section in the manuscript
Thank you for this comment. We agree that there are many hypotheses that explain the mechanism of treatment resistance that are not covered in this review. The purpose of this statement was to emphasize the importance of developing treatments that target different pathological processes and not to explain the mechanism of treatment resistance. We have rephrased the statement accordingly.
Comment: in the abstract, line 11, please remove the word "treatment" as this is a repetition
Response: Thank you, this has been corrected.
Comment in line 18, please list also other mood disorders as reported in DSM, or rephrase, since MDD and BD are the two most common but not the only mood disorders.
Response: Thank you for this comment. The focus of our review was on MDD and BD being the two most common mood disorders. We have removed the word “mood disorder” from line 18.
Comment: in line 19, the word "inflicted" does not sound appropriate
Response: Thank you for this comment, as suggested we have rephrased this sentence.
Comment: in line 31, authors define treatment resistance, but only for MDD and not for BD. Please see also my previous comments regarding this issue.
Response: Thank you for this comment. We have removed the definition of treatment resistance for MDD to resolve this comment.
Comment: in line 47, authors state that relevant literature was obtained: please define if relevant refers to quality or congruency with the topic
Response: Thank you for this comment, we have adjusted this to reflect congruency with the topic.
Comment: similarly, in line 57, please describe how high quality was defined for meta-analyses
Response: When a significant number of RCTs were available for one specific class of metbolic-targeting medication, we reviewed recent meta-analyses from reputable peer-reviewed journals that followed standard (e.g. PRISMA) reporting guidelines.
Comment: in line 185, authors report previous findings for TZD and other disorders, but neurological would sound more proper than neuropsychiatric, since findings only relate to Alzheimer and multiple sclerosis
Response: Thank you for this comment, we have adjusted this wording.
Comment: in line 294, please report PFC and HC in full
Response: Thank you, we have defined these acronyms.
Reviewer 2 Report
In the present review the Authors aimed to synthesize current knowledge on the association between mood disorders and metabolic dysfunction. They also summarized the current evidences on the efficacy of pharmacotherapies that target metabolic dysfunction on improvement of mood symptoms.
Overall, I found the present review concise, straightforward, timely, well conducted, very interesting and scientifically sound: enjoyed reading it! I have only some minor comments aimed to improve the high quality of the paper and these are outlined below:
1) Concerning lipids levels, several studies have pointed out a relationships between alexithymia, depression and serum lipid levels. This point shoud be briefly discussed (see De Berardis et al. J Biol Regul Homeost Agents. 2009 Jul-Sep;23(3):133-40).
2) Is it possible that targeting inflammation in both MDD and BD may lead to an improvement of metabolic parameters? Please briefly comment on this point.
Author Response
Comment: In the present review the Authors aimed to synthesize current knowledge on the association between mood disorders and metabolic dysfunction. They also summarized the current evidences on the efficacy of pharmacotherapies that target metabolic dysfunction on improvement of mood symptoms. Overall, I found the present review concise, straightforward, timely, well conducted, very interesting and scientifically sound: enjoyed reading it!
Response: Thank you for your time and feedback in reviewing our manuscript. We are delighted you enjoyed reading it.
Comment: Concerning lipids levels, several studies have pointed out a relationship between alexithymia, depression and serum lipid levels. This point shoud be briefly discussed (see De Berardis et al. J Biol Regul Homeost Agents. 2009 Jul-Sep;23(3):133-40).
Response: Thank you very much for this comment. We have included a comment of the association between alexithymia, suicidality, and cholesterol levels in the lipid regulation section (Page 4)
Comment: Is it possible that targeting inflammation in both MDD and BD may lead to an improvement of metabolic parameters? Please briefly comment on this point.
Response: Thank you very much for this comment. We agree with the reviewer that a reactive inflammatory system may be contributing to the association of metabolic disturbances of mood disorders. In the revised Discussion section we have included and expanded on this association. We have also included a comment on the anti-inflammatory effects of lipid modulating agents. In addition, we have proposed that future work should explore if prospectively targeting inflammation in both MDD and BD may lead to an improvement in metabolic parameters.
Reviewer 3 Report
Tables should be improved, i.e RCT are not sufficiently described, almost no information on the population is provided. Please provide type of the population, sex, age, and better specify design of the study (parallel or crossover, double blind, time of follow up, etc.)
Discussion should be more elaborated by providing a better overview of the mechanisms underlying the link between explored factors and brain disorders.
Author Response
Comment: Tables should be improved, i.e RCT are not sufficiently described, almost no information on the population is provided. Please provide type of the population, sex, age, and better specify design of the study (parallel or crossover, double blind, time of follow up, etc.)
Response: Thank you for this comment. We have updated the tables to include additional information on the population and study design of the included studys.
Comment: Discussion should be more elaborated by providing a better overview of the mechanisms underlying the link between explored factors and brain disorders.
Thank you very much for this comment. We have included a section in the revised Discussion that discusses potential mechanisms underlying the link between metabolic disturbances and mood disorders.
Reviewer 4 Report
The authors need to strengthen the biological basis between mood disorders and metabolic changes.
Author Response
Comment: The authors need to strengthen the biological basis between mood disorders and metabolic changes
Response: Thank you to the reviewer for this comment. We have included a section in the revised Discussion that discusses potential mechanisms underlying the link between metabolic disturbances and mood disorders.
Reviewer 5 Report
Generally a very well thought out, pleasant to read article which will benefit future studies regarding metabolic dysfunction in mood disorders. Some interesting aspects can also be found in this relatively recent article: doi: 10.4183/aeb.2019.342
Author Response
Comment: Generally, a very well thought out, pleasant to read article which will benefit future studies regarding metabolic dysfunction in mood disorders. Some interesting aspects can also be found in this relatively recent article: doi: 10.4183/aeb.2019.342
Response: Thank you very much for the kind words and review of our manuscript.
Round 2
Reviewer 1 Report
Authors addressed all the main issues raised by the referee.
Author Response
Thank you for reviewing our resubmitted manuscript
Reviewer 3 Report
Minor comments
Table 1. please substitute "&" with "and".
Please add table footnote with abbreviations.
The part describing mechanisms in the discussion section should be descriptive and not presented as bullet points.
Authors pointed out that gut microbiota may impact psychiatric disorders, but it was also shown that day may affect brain development, function and metabolism, please see: PMID: 30843443.
Author Response
Comment: Table 1. please substitute "&" with "and".
Response: Thank you, this has been revised
Comment: Please add table footnote with abbreviations.
Response: Thank you, abbreviations have been added to footnotes
Comment: The part describing mechanisms in the discussion section should be descriptive and not presented as bullet points.
Response: Thank you for this comment, we have removed the bullet points.
Comment: Authors pointed out that gut microbiota may impact psychiatric disorders, but it was also shown that day may affect brain development, function and metabolism, please see: PMID: 30843443.
Response: Thank you for this comment. We have made reference to this point in our discussion.